# Functionalization of Fluorine on the Surface of SnO_2_–Mg Nanocomposite as an Efficient Photocatalyst for Toxic Dye Degradation

**DOI:** 10.3390/nano13172494

**Published:** 2023-09-04

**Authors:** G. Velmurugan, R. Ganapathi Raman, P. Sivaprakash, A. Viji, Shin Hum Cho, Ikhyun Kim

**Affiliations:** 1Department of Physics, Noorul Islam Centre for Higher Education, Kumaracoil, Kanyakumari 629180, Tamil Nadu, India; velgi_81@yahoo.com; 2Department of Physics, Saveetha Engineering College (Autonomous), Chennai 602105, Tamil Nadu, India; 3Department of Mechanical Engineering, Keimyung University, Daegu 42601, Republic of Korea; siva.siva820@gmail.com; 4Department of Physics, Kongunadu College of Engineering and Technology, Thottiyam 621215, Tamil Nadu, India; vijiarangarasan@gmail.com; 5Department of Chemical Engineering, Keimyung University, Daegu 42601, Republic of Korea; shinhum@kmu.kr

**Keywords:** SnO_2_–Mg–F nanoparticles, sol-gel method, methyl orange dye, safranin dye, photocatalysis

## Abstract

This work reports on the photocatalytic activity of tin oxide (SnO_2_)-doped magnesium (Mg) and fluorine (F) nanoparticles for methyl orange and safranin dye degradation under sunlight irradiation. Nanocatalysis-induced dye degradation was examined using UV–visible spectroscopy and a pseudo-first-order kinetics model. The results indicate that the prepared nanoparticles exhibit superior photocatalytic activity, and the degradation of methyl orange (MO) dye is approximately 82%. In contrast, the degradation of safranin dye is 96% in the same time interval of 105 min. The calculated crystallite size of the SnO_2_–Mg–F nanocomposite is 29.5 nm, which respects the particle size found in the DLS analysis with a tetragonal structure and spherical morphology affirmed. The optical characteristics were assessed, and their respective bandgap energies were determined to be 3.6 eV. The influence of F in Mg and SnO_2_ is recognized with the XRD and FT-IR spectra of the prepared particles.

## 1. Introduction

The lack of access to purified water for a large population is only one of the problems that environmental contamination on a global scale is causing [1]. Materials science has become more interested in advancing environmental remedial technologies for environmentally friendly energy resources, focusing on water [2]. These worldwide issues may be mitigated using conventional treatments, including ion exchange, carbon filtering, biological processes, and reverse osmosis [3]. These techniques have several drawbacks, including high costs, complexity, and energy use. One of the most effective, innovative, environmentally friendly technologies for purifying water is photocatalytic degradation. In the presence of a photocatalyst, the organic contaminants are quickly and efficiently broken down through photocatalytic oxidation–reduction processes. Uncontaminated water is environmentally hazardous because of the organic dyes used in many sectors [4].

Researchers have shown great interest in semiconductor nanoparticles over the last several decades, owing to their exclusive physicochemical properties, such as their non-toxicity and environmental sustainability [5]. Therefore, SnO_2_ has attracted recognition as a potent photocatalyst. While SnO_2_ shows promising photocatalytic activity, its performance is limited by the rapid recombination of photogenerated electron-hole pairs [6]. It is desirable for catalysis because of its transparency, electrical conductivity, and chemical stability [7]. Poor quantum yields due to the quick recombination rate and inadequate exploitation of light-produced electron-hole pairs remain a constraint in the photocatalytic efficacy of tin oxide nanocatalysts required to satisfy the needs of practical applications [8]. Nanostructure, composition, and surface modification of semiconducting materials have been demonstrated in recent years to impact their photocatalytic capabilities significantly [9]. The photocatalytic performance of tin oxide nanocatalysts may be enhanced in two ways. One method involves controlling the particle size and morphology of SnO_2_ to obtain a sizeable adequate quantum size and a high specific surface area [10]. Another method is to dope SnO_2_ with additional elements, like noble metals, metal ions, or semiconductors, to function as traps for photo-induced holes and electrons, enhancing the material’s photocatalytic efficacy [11].

The seventh furthermost prevalent element on Earth, magnesium (Mg), consumes many desirable characteristics [12]. It is safe to handle, inexpensive compared to other metals, recyclable, and simple to work with regarding metallurgy [13]. As a result, there has been much focus on materials made from magnesium. Biochemical sensing, nanoscale electronics, photocatalytic activity, and optoelectronics are the only possible uses of Mg-doped n-type semiconducting oxide materials [14]. Since the ionic radius of Mg is similar to that of Sn, it follows that Mg doping may be predicted to replace the Sn host lattice, lowering the energy of the band gap and, in turn, boosting the photocatalytic performance [15].

Concerning other dopants, fluorine is among the most advantageous anionic impurities for SnO_2_. Since the ionic radius of F^−^ is equivalent to that of O_2_^−^, the Sn–F bond corresponds to the Sn–O bond. SnO_2_ surface adsorption characteristics are drastically altered by the adsorption of fluoride ions, which also dramatically enhance surface hole transfer and the preferred creation of hydroxyl radicals (OH*) with higher reactivity [16]. Tuning the band structure and modifying localized electronic structures are two more effective methods for enhancing photo-reactivity via doping [17].

Metal oxides, such as zinc oxide [18], titanium oxide [19], and nickel [20], are efficient in photocatalysis as a single element. Still, many attempts have been conducted for the doping of various metal oxides, such as iron [21], cobalt [22], and vanadium [23], aiming to extend the photo response from the SnO_2_ from ultraviolet to the visible region. Among the metal oxide composites, tin-doped magnesium has a better photocatalytic performance. On the other hand, doping non-metal elements with metal nanoparticles could be a possible approach to improve the photocatalytic activity of the nanocomposite or nanoparticles [24].

Beyond that, ion doping could be another effective means to enhance the electrochemical performance of SnO_2_ nanoparticles. In this respect, it is worthwhile to note that the conductivity and electrochemical reversibility of SnO_2_ can be significantly improved by doping with nonmetal ions, such as Cl, S, P, or F atoms [25]. Due to the similar ionic size of F^−^ and O^2−^, F has been regarded as the most efficient doping agent for SnO_2_ among these dopants. Hence, F-doped nanoparticles have gained increased interest and have been extensively investigated for their diverse applications, such as gas sensors, transparent conductors, and photocatalysts [26]. However, the performance of the F-doped SnO_2_–Mg composite has not been well studied for photocatalytic activity. Therefore, we expect to combine the advantages of F doping and metal ion (Mg) incorporation so that the composite material could exhibit outstanding photocatalytic dye degradation performance.

Thus, in our work, SnO_2_ was doped with magnesium (Mg) and fluorine (F) through a simple sol-gel technique [27]. Doping with metals has improved the separation rate of photogenerated electron-hole pairs. The efficiency of the doped SnO_2_ is assessed on commercially available dyes like methyl orange and safranin dyes [28]. The physico-chemical properties of the synthesized samples are investigated. This is the first preliminary study of the structural, optical, and photocatalytic properties of SnO_2_–Mg–F nanoparticles.

## 2. Experimental Procedure

### 2.1. Materials

Stannous chloride (SnCl_4_), ammonium fluoride, and magnesium chloride were employed to create a title nanocomposite as an organic pollutant for photodegradation. Saffranine and methyl orange dyes were utilized. All of the compounds described here were reagent grade and purchased from Aldrich. All solutions were prepared using distilled water. For the metal precursor, a homogeneous solution combination of 1 mole reagent-grade stannous (IV) chloride (SnCl_4_, >99.9% purity, molecular weight 189.68), 1 mole ammonium fluoride, and 1 mole magnesium chloride (Sigma-Aldrich, St. Louis, MO, USA) was made.

### 2.2. Preparation of SnO_2_–Mg–F Nanocomposites

To make the SnO_2_–Mg–F nanocomposite, 1 M of tin chloride, magnesium chloride, and ammonium fluoride was placed in 150 mL of distilled water. Then, the solution was stirred vigorously until complete dissolution of the precursor. After that, until the pH of the mixture was 10, sodium hydroxide solution was introduced and stirred for 3 h. The white precipitate was washed using water and ethanol. Then, the cleaned samples were dried using a hot air oven. The product was then calcined for five hours in an air atmosphere in a muffle furnace at 550°. Schematic representation for the synthesis of the SnO_2_–Mg–F nanocomposite is shown in Figure 1.

### 2.3. Sample Preparation for Decolorisation

The photocatalytic performance of the SnO_2_–Mg–F nanocomposite was carried out in cationic (safranin) and anionic (methyl orange) dyes under sunlight. Then, 100 mg of the prepared nanocomposite was added to 1000 mL of the prepared dye solutions. The nanocomposite-suspended dye was irradiated under sunlight for 105 min with an interval of 15 min, where the appropriate amount of dye solution was subjected to UV-Vis spectrophotometer analysis. Then, the UV-visible spectrum was recorded, and the dye degradation depended on the optical absorption level. The efficiency of dye degradation was calculated using the equation [29],
$$\mathrm{Dye}\;\mathrm{degradation}\;\mathrm{efficiency}=\left(1-\frac{C_{t}}{C_{0}}\right)\times 100\%$$
where C_t_ and C_0_ indicate the final and initial absorbance of safranine and methyl orange.

### 2.4. Characterization Techniques

XRD (X’Pert PRO; PANalytical, Almelo, the Netherlands) was used at 40 kV and 30 mA with a Cu anode to seek the phases of NPs. The study was performed in the 2θ range of 20–80 with a step size of 0.0170. To analyze the functional groups, present in the proximity of wavelengths of 4000 to 400 cm^−1^, Fourier transform infrared spectroscopy (FT-IR) (Spectrum 100; PerkinElmer, Waltham, MA, USA) investigation was executed with KBr as the carrier in a ratio of 1:200. With the dynamic lighting effect method, the average particle size was studied using a particle size analyzer (Nanophox; Sympatec, Clausthal-Zellerfeld, Germany) with a laser beam at a wavelength of 633 nm. The process was repeated several times in the 1–1000 nm range to obtain an average size of the particles. The absorbance of the material was measured using UV-Vis spectroscopy (Shimadzu UV-Visible spectrophotometer, model UV 1800, Shimadzu, Tokyo, Japan) at a wavelength of approximately 180–800 nm with double-distilled water as a background. The material’s bandgap was calculated using the UV-Vis spectroscopy results. The emission properties of the prepared material were studied using photoluminescence spectroscopy (Cary Eclipse, Agilent Technologies, Singapore). The scanning electron microscopy (SEM) photographs were taken with a JEOL (Tokyo, Japan) device with a Gatan Quantum ER 965 imaging filter.

## 3. Results and Discussion

### 3.1. X-ray Diffraction Spectroscopy

The crystalline properties of the nanocomposite are shown in Figure 2. The results reveal the tetragonal structural phase of SnO_2_ by P42nmn space groups with the 2θ values of 26.71, 33.96, 38.06, 45.54, 51.9, 54.86, 66.17, and 72.32°, which corresponds to the JCPDS No 77-0452 [30]. Likewise, MgO shows the cubic structure by the space group of Fm3m with the 2θ values of 42.88, 62.04, and 78.77°, corresponding to the JCPDS No. 75-0447 [31]. Thus, the nanocomposite shows the mixed crystalline phase of both tin and magnesium without any other impurities. Initially, Sn^2+^ was first bonded with Mg ions through electrostatic force and then nucleated and grew into SnO_2_–Mg nanoparticles during the synthesis process. Finally, these nanoparticles were uniformly anchored by F ions on the surfaces of the SnO_2_–Mg nanoparticles to form the SnO_2_–Mg–F nanocomposites. As follows, to determine the crystallite size, the Debye–Scherrer equation was used [32],
$$D=\frac{K\lambda }{\beta COS\theta }$$
where k is constant (0.9), is the Cu Kα source’s wavelength with a value of 1.540, is the Bragg angle, and is the full width of the half maximum. The calculated crystallite size of the SnO_2_–Mg–F nanocomposite is 29.5 nm, which respects the particle size found in the DLS analysis. The length of the dislocation lines per unit volume or the dislocation density (δ) was calculated using the equation [33],
δ = 1/D^2^
where δ is the calculation of the number of defects in the crystal. The calculated dislocation density of the nanocomposite is 11.49 × 10^−3^, and the computed micro-strain is 0.0032%. The fluorine forms more bonds with the surface of other elements due to its high electronegativity and makes the composite highly crystalline [34].

The calculated structural parameters are slightly varied from the pure SnO_2_ nanoparticles. The variation in the structural parameter is highly based on the type of dopant incorporated with pure SnO_2_ nanoparticles. In most conditions, the dopants are supposed to increase the structural parameters, like crystal size, lattice parameters, volume, density, etc. [35].

### 3.2. Scanning Electron Microscope

The surface morphology of the SnO_2_–Mg–F nanocomposite is shown in Figure 3. The result confirms the formation of monodispersed spherical morphology, where the particles are slightly agglomerated due to the high surface energy [36]. The morphology possesses a uniform shape dense and porous nature. The results correspond to the DLS measurement, where the average particle size is approximately 132 nm, which satisfactorily matches the crystal size of the composite. The mono dispersion may be due to the addition of fluorine at the surface of the nanocomposite [37].

### 3.3. Functional Group Analysis

Figure 4 shows the SnO_2_–Mg–F nanoparticle FT-IR spectra in the 4000–400 cm^−1^ wavenumber. Several spectral bands were created by the physical adsorption of water by the nanoparticles; these bands correspond to O-H stretching at approximately 3438 cm^−1^ and H-OH bending at approximately 1632 cm^−1^ [38]. Both a C-H stretching band at approximately 2922 cm^−1^ and a C-H bending band at approximately 2852 cm^−1^ occur. The band between 400 and 600 cm^−1^ appears to be the stretch vibrations in tin oxide (Sn–O). An F–F bond is responsible for the appearance of the peak at 901 cm^−1^ [39]. The Sn–O and O–Sn–O stretching vibrations are responsible for the prominent peaks at approximately 600–650 cm^−1^ [40]. Carboxylic acid’s OH bending is responsible for the 1438 cm^−1^ frequency. The vibration of Mg–O is responsible for the strong and wide band at 477 cm^−1^ [41]. 

### 3.4. Particle Size Analysis

The dynamic light scattering technique for SnO_2_–Mg–F is shown in Figure 5. The average particle (d_50_) is 122 nm, also seen in the SEM images. The broadened peak indicates that the maximum particles have the same particle size at d_50_ [42]. The size of the particle at its maximum (d_90_) and minimum (d_10_) is 14 and 1447 nm, respectively. The increased particle size at d_90_ is due to forming a composite of three components. Meanwhile, fluorine at the material’s surface maintains the constant d_50_ value and shows broad peaks.

### 3.5. UV-Visible Spectroscopy

The optical absorption spectrum of SnO_2_–Mg–F is displayed in Figure 6a. It is identified that the prepared nanocomposite maintained an absorption band at 235 nm in the UV region. The charge transfer shift between the O (2p) and Sn (4d) states in O_2_^−^ and Sn^4+^ is responsible for the UV absorption band of SnO_2_ [43]. Meanwhile, MgO does not show any sharp peak, indicating that nanoparticles with different sizes are produced in this method [44], which is evidenced. The doping level is the main factor responsible for the prepared catalyst’s optical absorption. The F-doped SnO_2_–Mg nanoparticles exhibited stronger absorption in the UV-Vis range than pure SnO_2_ with a red shift in the bandgap transition [45]. The F dopants mainly exist and stick on the nanoparticles surface, which enhances the light absorption and improves the photocatalytic performance. The band gap of the SnO_2_–Mg–F nanoparticles was calculated using Tauc’s equation [46];
$$(\alpha \;h\nu )=A\;{\left(h\nu -Eg\right)}^{n}$$
where *A* is constant and has varied with transitions, *hν* is the incident photon energy, *Eg* is the energy gap, *α* is the absorption coefficient, and *h* is the Planck constant. The exponent value *n* represents the nature of the electronic transition as direct or indirect. The calculated band gap was 3.6 eV, as shown in Figure 6b, improving the photocatalytic performance efficiency. The nanocomposite has a transmittance of 62%.

### 3.6. Photoluminescence Spectroscopy

The luminescence properties of the SnO_2_–Mg–F nanocomposite between 200 and 600 nm are depicted in Figure 7. The nanocomposite exhibits an excitation at 257 nm, showing multiple peaks at the emission spectra. The 361 and 484 nm peaks correspond to SnO_2,_ whereas the 424 and 459 nm peaks are attributed to MgO. The initial peak at 361 nm (UV emission) is due to the interband radiation combination of photo-generated electrons and holes [47]. The emission at 484 nm intimates the blue-green emission, which occurs due to the transition mediated by defect levels in the band gap and corresponds to oxygen vacancies on the surface [48]. The peak at 424 nm is reveling the band-to-band shift in MgO. The 481 emission is due to the recombination of electrons with oxygen vacancies [49]. Finally, the intense peak at 528 nm is due to the addition of fluorine, which shifts the emission toward blue [50]. Multiple peaks indicate the polycrystalline nature of the material [51].

### 3.7. XPS Analysis

X-ray photon spectroscopy determines the synthesized sample’s surface composite elements, purity, and chemical state. The peaks found on the survey spectrum in Figure 8 of the SnO_2_–Mg–F nanocomposite show mainly Mg, Sn, F, and O, confirming the high purity of the SnO_2_–Mg–F nanocomposite. Figure 8b shows that the Mg 1s peak deconvoluted at 1304.06 eV is assigned to MgO with Mg^2+^ valence state in the SnO_2_–Mg–F nanocomposite, and it is found to be matching with the literature value for MgO nanoparticles, and the 1305.98 eV peak corresponds to the Mg–OH. The O 1s peak for the SnO_2_–Mg–F nanocomposite can be deconvoluted into three peaks, as shown in Figure 8c. The lower binding energy peak at 530.08 eV and 531.9 eV is due to the lattice oxygen of the SnO_2_–Mg–F nanocomposite, whereas the higher binding energy peak at 534.49 eV is due to absorbed oxygen molecules on the surface of the nanocomposite [52].

The F 1s spectrum was fitted at 685.6 eV and 687.5 eV. Correspondingly, the component at 685.6 eV is related to the fluorine of the Sn–F bond in the SnO_2_–Mg–F nanocomposite; finally, the component at 687.5 eV is related to the F ions that form hydrogen bonds with the surface hydroxyl groups [53]. Figure 8e shows the Sn 3d spectrum from which the peaks for Sn are seen at 486.4 eV and 494.91 eV, which is due to Sn 3d5/2 and 3d3/2 corresponding to the standard data for SnO_2_, and the satellite peaks at 488.46 eV and 496.9 eV correspond to Sn 3d5/2 and Sn 3d3/2. The spin-orbit splitting of 8.51 eV shows the presence of Sn^4+^ in SnO_2_. Sn^+2^ or Sn^+0^ is not seen in the spectrum [54].

### 3.8. Photocatalytic Activity

UV-visible spectroscopy was used to examine a blank dye (methyl orange and safranin) control experiment to assess whether a reaction occurred [55]. After measuring the UV-visible of the blank dye, the organic dye solution was applied to the developed NPs, and spectra were obtained. The optical absorption values of the pure anionic dye (methyl orange) and cationic dye (safranin dye) are initially measured at 464 and 519 nm using UV-Vis spectra without a nanocatalyst. The time-dependent photocatalytic activity of the SnO_2_–Mg–F nanoparticles was analyzed for dye degradation at 15 min intervals for up to 105 min. When the irradiation duration increases from 0 to 105 min, the suspended dye solutions’ vigorous absorption/peak intensity decreases with time.

Further, it is evident that the nanocatalyst influences the gradual decolorization of the dye solution and varies with irradiation time. The results in Figure 9 and Figure 10 clearly show that, under sunlight irradiation, the photocatalytic performance of pure SnO_2_ is greatly improved with the doping of Mg and co-doping of F ions. The improved absorption could be attributed to the narrowing of the SnO_2_ bandgap, which enhances the light absorbance. On the other hand, the significant enhancement could also be due to the synergistic effect between Mg and F atoms with SnO_2_. Co-doping with magnesium and fluorine could prevent the recombination of electron-hole pairs, thereby improving the photocatalytic performance [56].

In the presence of the SnO_2_–Mg–F nanoparticles as photocatalysts, the absorption spectra of safranin dye and methyl orange modify, as shown in Figure 8 and Figure 9. Also, the photodegradation rates (C/Co) of safranin dye and methyl orange in the presence of catalysts when exposed to sunlight are shown in Figure 9 and Figure 10c. Figure 9 and Figure 10b summarize the time-dependent efficiency of pure dye degradation with and without catalysts using SnO_2_–Mg–F NPs exposed to sunlight. After 105 min of irradiation, the degradation effectiveness of the SnO_2_–Mg–F nanoparticles is 82% for methyl orange and 96% for safranin dye, respectively. Therefore, the photocatalytic efficiency is more significant for the cationic safranin dye than the anionic methyl orange dye. The higher degradation efficiency could be due to the surface charge of the nanocatalyst, which is negative as per previous reports [57]. Introducing fluorine increases the nanocatalyst’s negative surface charge due to high electronegativity. The negatively charged nanocatalyst attracts the positively charged saffranine dye and shows maximum degradation [58].

The reactive kinetic performance of the obtained samples was carefully investigated to study their photocatalytic characteristics. The safranin dye and methyl orange photodegradation rates were calculated using the equation below, which was modified using the pseudo-first-order kinetic process [59].
$$Kt=ln\frac{C}{Co}$$

The reaction concentration is augmented as C/Co, where Co is the initial concentration, and C is the methyl orange and safranin dye concentrations after the degradation time.

The kinetic degradation rate constant is shown in Figure 8 and Figure 9d, and the obtained ideals are 0.01474 min^−1^ for methyl orange and 0.03133 min^−1^ for safranin dyes, and 0.00017 and 0.00015 min^−1^ for methyl orange and safranin dyes, individually, without a catalyst.

The detailed photocatalytic mechanism of the prepared photocatalyst is shown in Figure 11. The photoexcitation of the nanoparticles, which forms electron-hole pairs on the catalyst’s surface, initiates the breakdown of the methyl orange and safranin dyes in aqueous solutions. Furthermore, due to oxygen vacancies, our study boosted the photocatalytic activity of the synthesized nanoparticles [60]. According to the results, the synthesized SnO_2_–Mg–F NPs outperform the cationic and anionic dyes as a photocatalyst when exposed to sunlight. Because the light intensity produced by solar energy is much higher, energy absorption and electron migration from the valence band to the conduction band occur, boosting nanoparticle photocatalytic activity [61]. Photoexcited holes (h+) may react with oxygen molecules to generate H_2_O and hydroxyl (*OH), whereas photoexcited electrons (e^−^) can be trapped via oxygen molecules to form (*O_2_) radicals. SnO_2_ quickly absorbs these photoexcited (e^−^) molecules [62]. The released (e) from SnO_2_’s surface is attracted by the partially attenuated oxygen molecules in water, where they react to form superoxide radical anions (*O_2_) and hydroxyl radicals (*OH) [63].
SnO_2_–Mg–F nanoparticles + Dye + Visible light → h^+^ (VB) + e^−^ (CB)
Dye molecule + OH → *OH^−^ + Degradation Products
Dye molecule + O_2_ → *O^−^_2_ + Degradation Products

The synergistic effects of co-doping with Mg and F are the main reasons for the enhanced activity of the doped SnO_2_. Introducing F dopants into Mg-doped SnO_2_ further narrows the SnO_2_ bandgap, promoting visible light absorption [64]. The promoted light absorption enhances the production of photoelectrons and holes, the primary carriers for photocatalytic activity. On the other hand, the doped F^2+^ is located on the surface of SnO_2_, where the reconstruction enhances the photocatalytic activity of the Mg-doped SnO_2_ under the irradiation of visible light. The fluorine dopant does not contribute to the visible light absorption of SnO_2_ [65]. However, F dopant promotes the creation of oxygen vacancies, promoting light absorption in the visible-light region and producing oxidizing species that enhance the photocatalytic activity [66,67]. The co-doping with Mg and F improved the photocatalytic efficiency of SnO_2_ through narrowed bandgap, promoted the separation of photo-generated electrons and holes, and enhanced photocatalytic oxidizing species. From the earlier reports, the comparison assessment of the photocatalytic activity of pure SnO_2_ and doped SnO_2_ nanoparticles is listed in Table 1.

In order to assess the repeatability of the catalytic sample reactions, the ability of the same catalysis process to degradation of a dye over the course of several days was evaluated. When the dates of the collected data are displayed, we can observe that, as the number of days rises, the sample’s reproducibility and stability are slightly varied as shown in Figure 12. Additionally, we observed from the study that the prepared material has good stability at different dyes for several days.

## 4. Conclusions

The SnO_2_–Mg–F nanocomposite was successfully prepared through a soft chemical method. The addition of fluorine significantly impacts the nanocomposite’s crystalline nature, particle size, and morphology. The nanocomposite shows a monodispersed spherical morphology with an average crystallite size of 29.5 nm and a particle size of 122 nm. The nanocomposite possesses better optical properties with a 3.6 eV band gap. This band gap affects the photocatalytic activity with 96% and 82% degradation for cationic and anionic dyes, respectively. The fluorine is highly responsible for high efficiency in cationic dyes, which induces a high negative charge over the surface of the nanocomposite.

## Figures and Tables

**Figure 1 nanomaterials-13-02494-f001:**
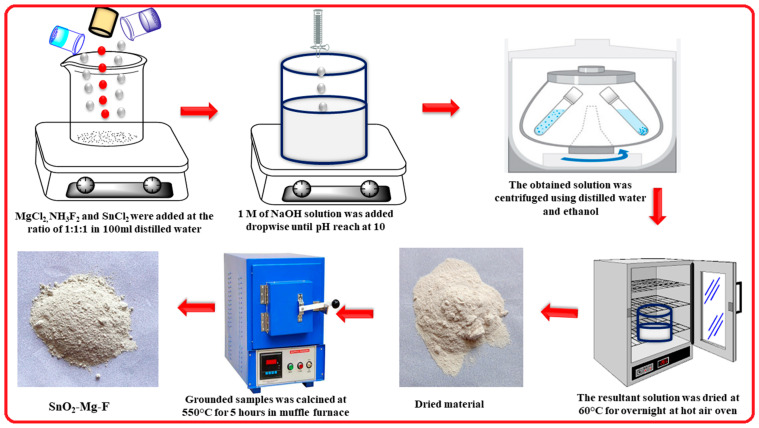
Schematic representation for the synthesis of the SnO_2_–Mg–F nanocomposite.

**Figure 2 nanomaterials-13-02494-f002:**
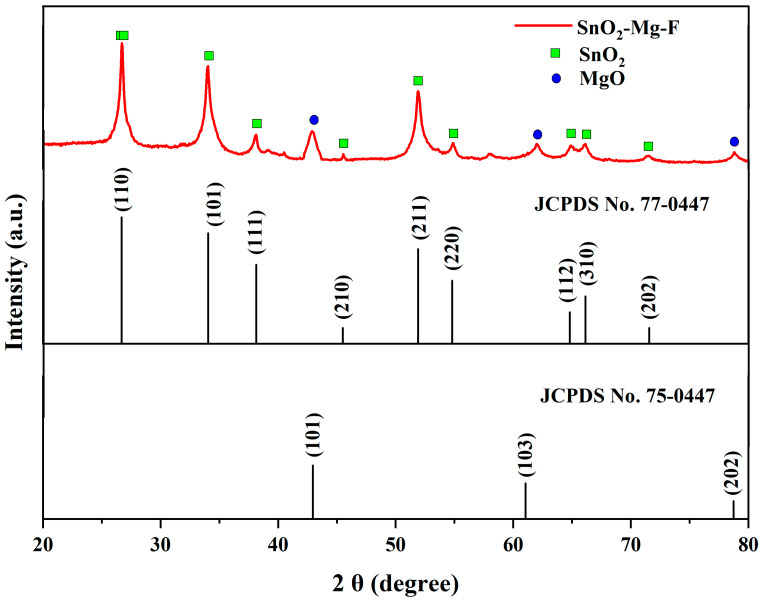
XRD analysis for the SnO_2_–Mg–F nanocomposite.

**Figure 3 nanomaterials-13-02494-f003:**
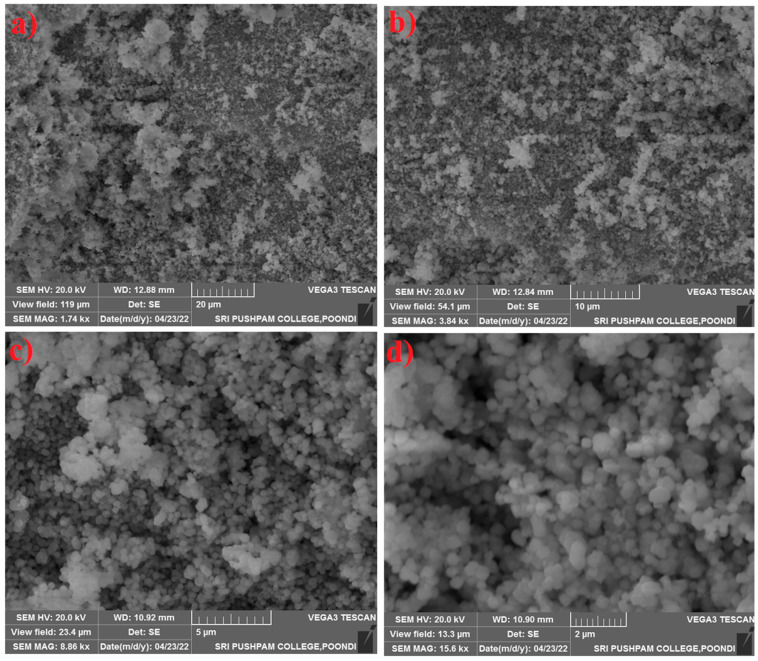
SEM images of the SnO_2_–Mg–F nanocomposite (**a**) 20 μm (**b**) 10 μm, (**c**) 5 μm, (**d**) 2 μm.

**Figure 4 nanomaterials-13-02494-f004:**
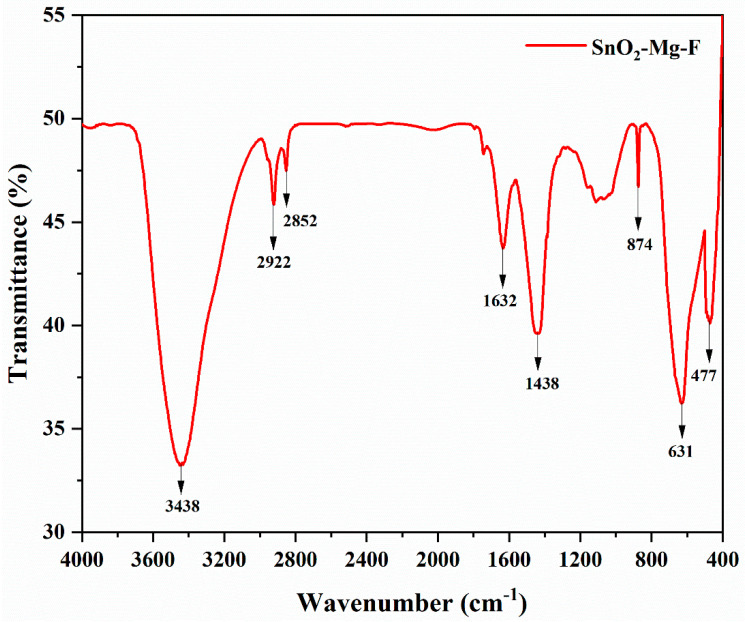
FT-IR spectra of the SnO_2_–Mg–F NPs.

**Figure 5 nanomaterials-13-02494-f005:**
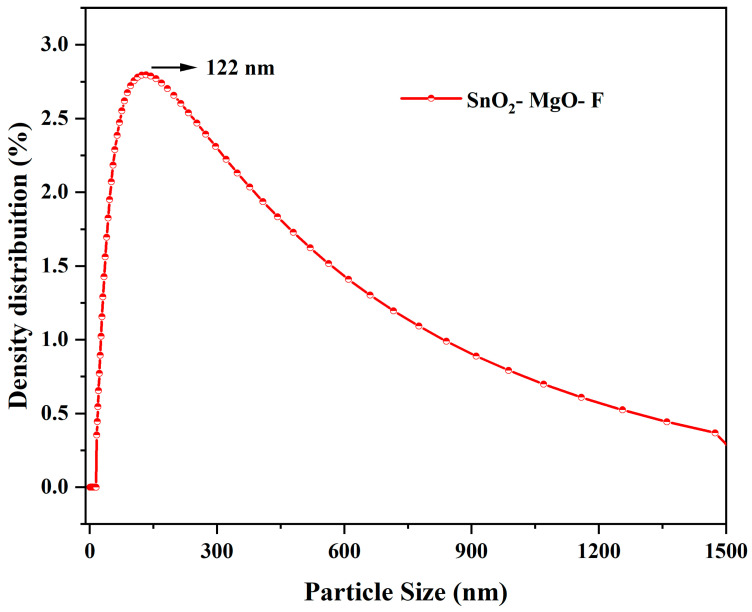
Particle size analysis for the SnO_2_–Mg–F nanocomposite.

**Figure 6 nanomaterials-13-02494-f006:**
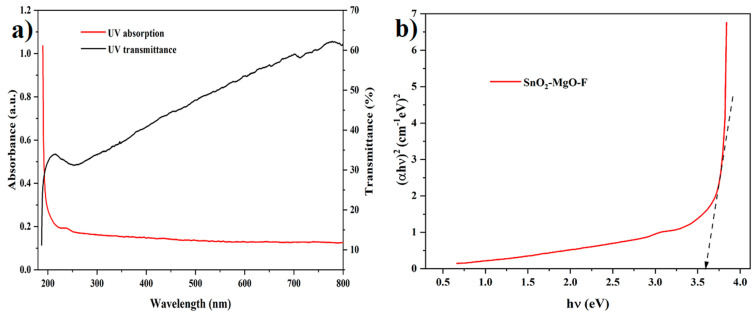
UV-Vis spectra and bandgap for the SnO_2_–Mg–F nanocomposite (**a**) absorption spectrum and (**b**) Photocatalytic performance efficiency of SnO_2_–Mg–F.

**Figure 7 nanomaterials-13-02494-f007:**
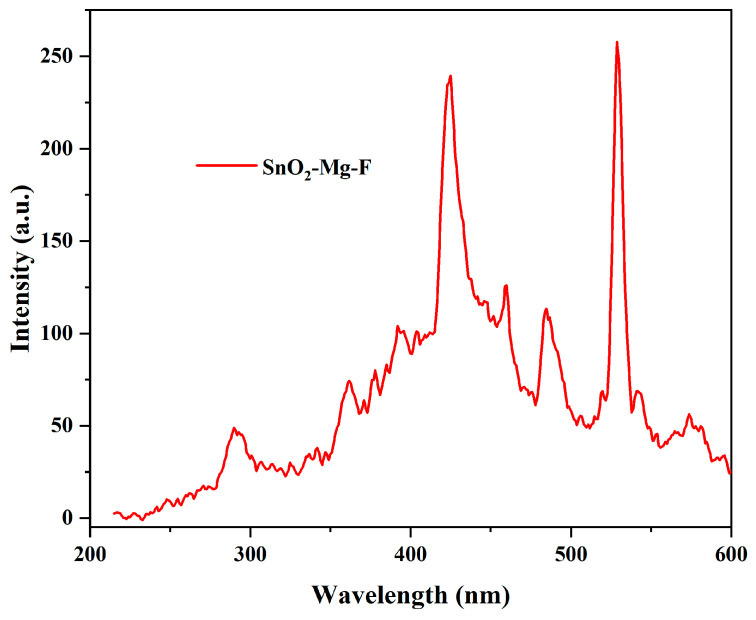
Photoluminescence spectra for the SnO_2_–Mg–F nanocomposite.

**Figure 8 nanomaterials-13-02494-f008:**
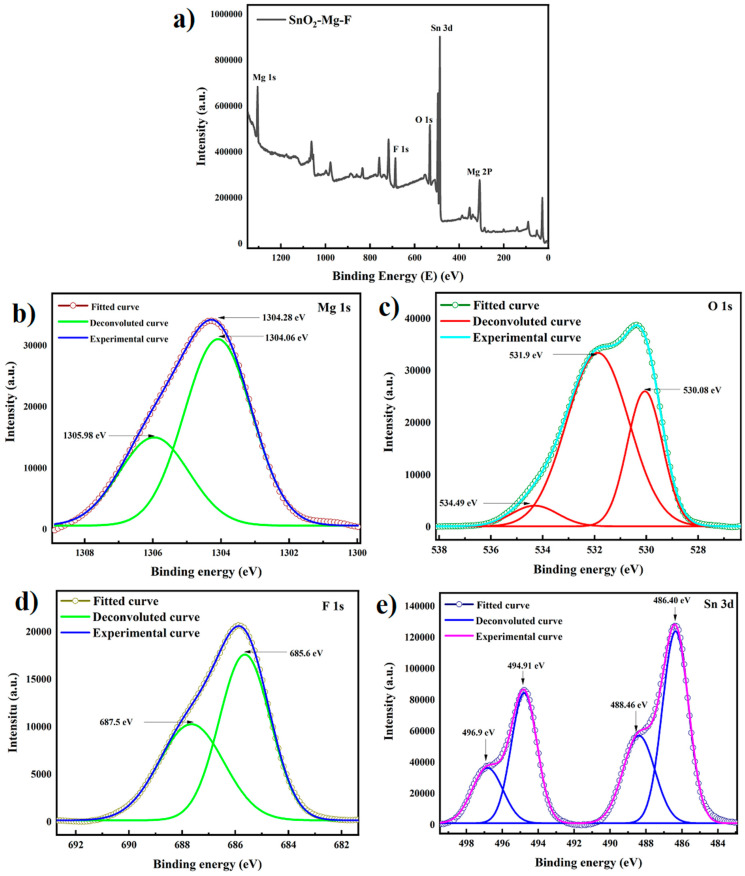
(**a**) XPS spectrum of the SnO_2_–Mg–F nanocomposite and XPS spectra of (**b**) Mg 1s core level, (**c**) O 1s core level, (**d**) F 1s core level, and (**e**) Sn 3d core level for the SnO_2_–Mg–F nanocomposite.

**Figure 9 nanomaterials-13-02494-f009:**
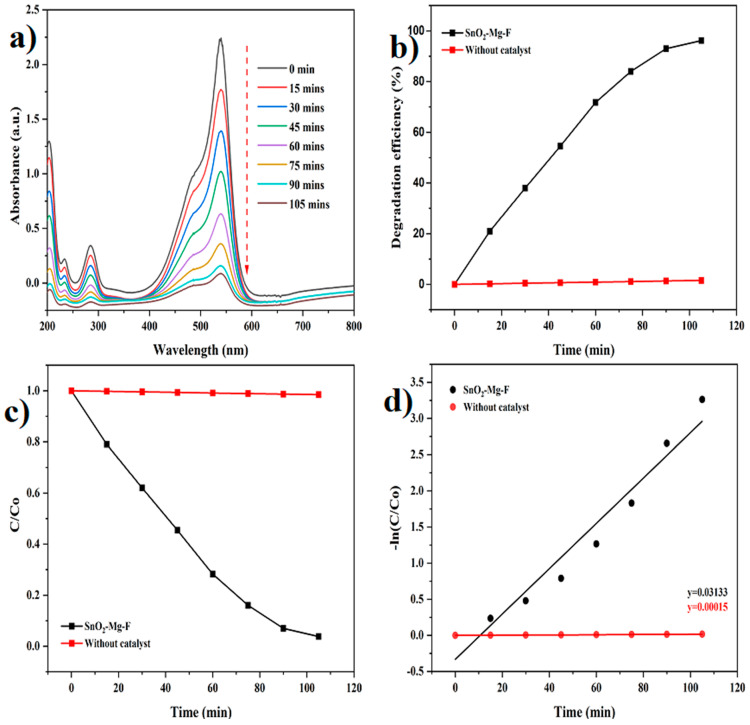
Photocatalytic degradation of safranin dye. (**a**) Degradation graph, (**b**) Degradation efficiency, (**c**) C/C_0_, (**d**) pseudo-first-order kinetics.

**Figure 10 nanomaterials-13-02494-f010:**
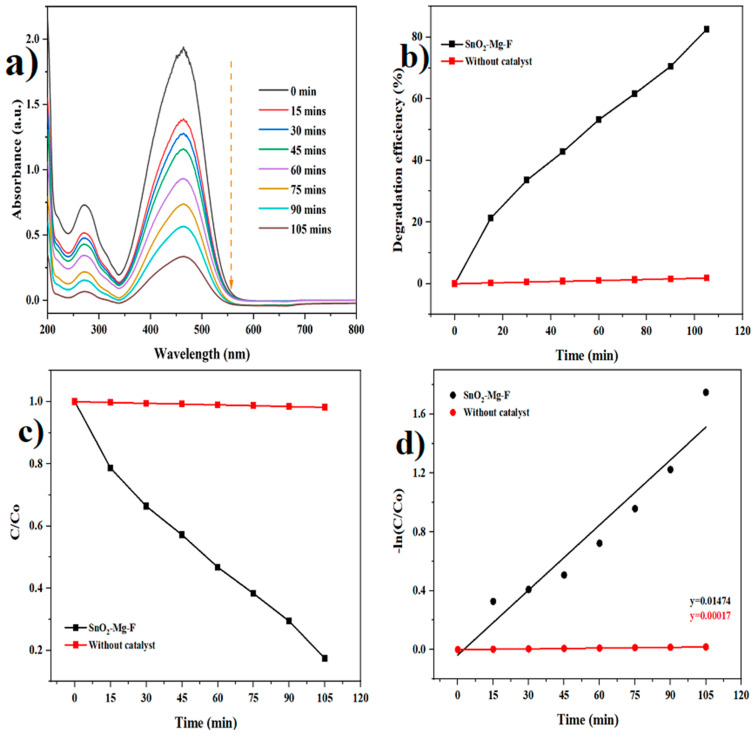
Photocatalytic degradation of methyl orange dye. (**a**) Degradation graph, (**b**) Degradation efficiency, (**c**) C/C_0_, (**d**) pseudo-first-order kinetics.

**Figure 11 nanomaterials-13-02494-f011:**
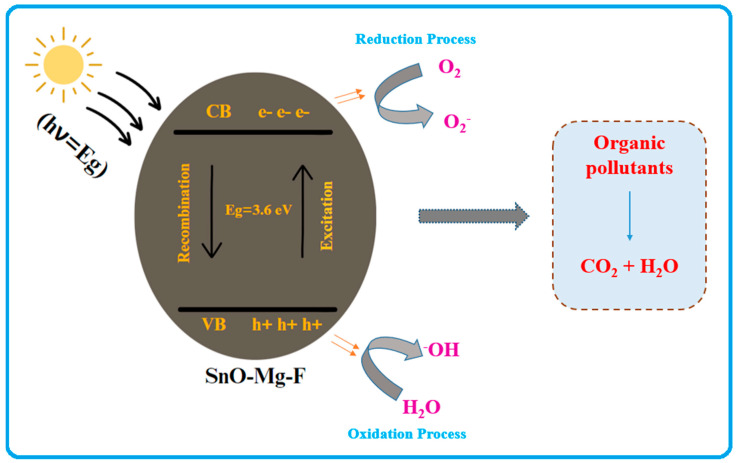
Mechanism of photocatalytic activity of the SnO_2_–Mg–F nanocomposite.

**Figure 12 nanomaterials-13-02494-f012:**
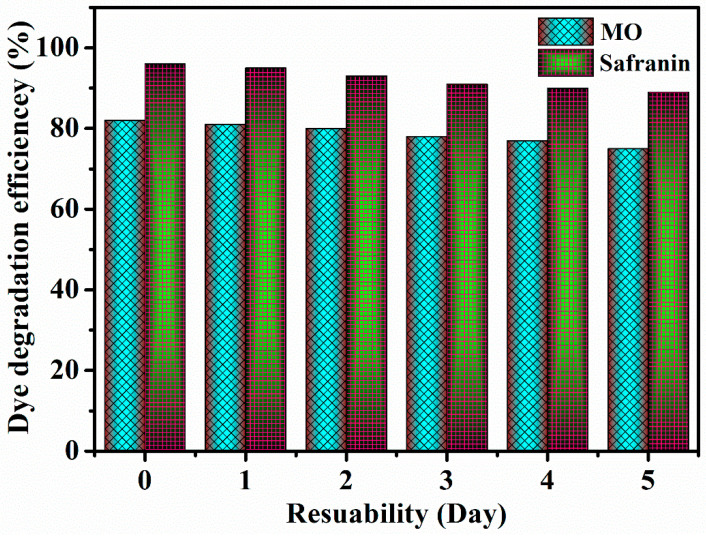
The repeatability of the catalytic sample reactions.

**Table 1 nanomaterials-13-02494-t001:** Comparison of the photocatalytic activity of pure SnO_2_ and doped SnO_2_ nanoparticles against various dyes.

Photocatalyst	Synthesis Method	Crystal Size (nm)	Bandgap (eV)	Dye	Reaction Time (min)	Degradation Efficiency (%)	Ref.
SnO_2_	Green synthesis	21	3.75	Methylene blue	120	89	[68]
SnO_2_	Green synthesis	7.6	3.8	Rhodamine B	150	92	[69]
SnO_2_	sol–gel	14.2	3.92	Methylene blue	90	93	[70]
SnO_2_	sol-gel	14.2	3.92	Methyl Orange	90	84	
ZnO–SnO_2_	Hydrothermal	30	3.4	Methylene blue	75	70	[71]
Sr–SnO_2_	Sol-gel	8.3	3.7	Methylene Orange	120	94	[72]
Mg–SnO_2_	Vapour pressure method	17.5	3.6	Alizarin Red S	120	75	[73]
F–TiO_2_	Sol-gel	18.79	3.3	Methylene blue	120	91	[74]
F–N–TiO_2_	Hydrothermal	20.6	3.5	Methylene blue	90	95	[75]

## Data Availability

Data and materials supporting the research are found within the manuscript. Raw data files will be provided by the corresponding author upon request.

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
