# Peer review of "Functionalization of Fluorine on the Surface of SnO2–Mg Nanocomposite as an Efficient Photocatalyst for Toxic Dye Degradation"

_nanomaterials, 2023, doi:10.3390/nano13172494_

Round 1
Reviewer 1 Report
The presented version of the manuscript was revised taking into account the comments of the reviewers. In particular, an additional study of the obtained samples of XPS spectroscopy methods was carried out, which will significantly improve the perception of this work not only by chemists and materials scientists, but also by physicists.
Author Response
Response to the reviewers’ comments
Title: Functionalisation of fluorine on the surface of SnO2-Mg nanocomposite as an efficient photocatalyst for toxic dye degradation.
Journal: Nanomaterials
Manuscript ID: Nanomaterials-2569068
We thank reviewer for evaluating the manuscript and providing valuable inputs for this manuscript. The detailed response to the pointed concern is as:
Reviewer #1:
In this work, the nanocomposite of SnO2 doped with Mg and F were successfully designed as a typical photocatalyst for dye degradation. According to the experimental characterization and reactive analysis, it was evident that the resultant dual dopants played the synergetic contributions for their superior efficiency. This is an alternative strategy for developing novel nanomaterials to couple with pollutant removal in environmental remediation. However, there are a few questions or concerns that must be fully addressed in the revised version.
The presented version of the manuscript was revised considering the comments of the reviewers. In particular, an additional study of the obtained samples of XPS spectroscopy methods was carried out, which will significantly improve the perception of this work not only by chemists and materials scientists but also by physicists.
Reply: Thanks for your valuable suggestions and comments.
We revised the paper as per the reviewer's suggestion and incorporated in the revised manuscript mention in the brown color. We believe that after introducing the inputs/suggestions by the reviewer, the manuscript has improved a lot and is now suitable for publication.
Reviewer 2 Report
The manuscript has been significantly improved, but further revisions are required according to my former comments, i.e., reactive species in reaction or reusability.
No
Author Response
Response to the reviewers’ comments
Title: Functionalisation of fluorine on the surface of SnO2-Mg nanocomposite as an efficient photocatalyst for toxic dye degradation.
Journal: Nanomaterials
Manuscript ID: Nanomaterials-2569068
We thank the reviewer for evaluating the manuscript and providing valuable input for this manuscript. The detailed response to the pointed concern is as:
Reviewer #2:
In this work, the results of the synthesis and characterization of the physical properties of a nanocomposite based on tin oxide (SnO2) doped with magnesium (Mg) and fluorine (F) atoms are presented, and the results of using this nanocomposite as a catalyst are analyzed, and its high photocatalytic activity and efficiency are shown. The work is a fully completed study. The content and conclusions correspond to the purpose of the experimental study indicated in the title.
Question 1: The manuscript has been significantly improved, but further revisions are required according to my former comments, i.e., reactive species in reaction or reusability.
Reply: Thanks for your valuable queries to improve the quality of the manuscript. In the future, we will carry out reusability studies.
We revised the paper as per the reviewer's suggestion and incorporated in the revised manuscript mention in the brown color. We believe that after introducing the inputs/suggestions by the reviewer, the manuscript has improved a lot and is now suitable for publication.
Round 2
Reviewer 2 Report
The paper was already improved, while some of my commens was not resolved, i.e., TOC data, reusability of catalyst.
No
Author Response
Response to the reviewers’ comments
Title: Functionalisation of fluorine on the surface of SnO2-Mg nanocomposite as an efficient photocatalyst for toxic dye degradation.
Journal: Nanomaterials
Manuscript ID: Nanomaterials-2569068
We thank the reviewer for evaluating the manuscript and providing valuable input for this manuscript. The detailed response to the pointed concern is as:
Reviewer #2:
Question: The paper was already improved, while some of my comments were not resolved, i.e., TOC data, reusability of catalyst.
Reply: Thanks for your valuable queries to improve the quality of the manuscript.
In order to assess the repeatability of catalytic sample reactions, the ability of the same catalysis process to degradation of a dye over the course of several days was evaluated. When the dates of the collected data are displayed, we can observe that as the number of days rises, the sample's reproducibility and stability are slightly varied as shown in Figure 12. Additionally, we observed from the study, the prepared material has good stability at different dyes for several days.
Regrettably, we must inform you that the TOC data you requested is not immediately available at our institution. We understand the importance of comprehensive data to support our findings, and we share your disappointment in not being able to fulfill this request at this time. But, we will do the measurement and report in future publications.
We revised the paper as per the reviewer's suggestion and incorporated it in the revised manuscript mentioned in yellow color. We believe that after introducing the inputs/suggestions by the reviewer, the manuscript has improved a lot and is now suitable for publication.

Round 3
Reviewer 2 Report
All my concerns were resolved. I suggest to accept the paper now.
No